# A Systematic Review of the Efficacy and Safety of Direct Oral Anticoagulants in Atrial Fibrillation Patients with Diabetes Using a Risk Index

**DOI:** 10.3390/jcm10132924

**Published:** 2021-06-29

**Authors:** Domenico Acanfora, Marco Matteo Ciccone, Valentina Carlomagno, Pietro Scicchitano, Chiara Acanfora, Alessandro Santo Bortone, Massimo Uguccioni, Gerardo Casucci

**Affiliations:** 1Unit of Internal Medicine, San Francesco Hospital, Viale Europa 21, 82037 Telese Terme, Italy; vale.carlomagno@gmail.com (V.C.); acanforachiara@gmail.com (C.A.); 2Section of Cardiovascular Diseases, Department of Emergency and Organ Transplantation, School of Medicine, University of Bari, Piazza Umberto I, 1, 70121 Bari, Italy; marcomatteo.ciccone@uniba.it (M.M.C.); piero.sc@hotmail.it (P.S.); 3Cardiology Unit, Hospital “F. Perinei”, Strada Statale 96 per Gravina in Puglia, 70022 Altamura, Italy; 4Department of Biotechnological and Applied Clinical Sciences, University of L’Aquila, 67100 L’Aquila, Italy; 5Division of Cardiac Surgery, Department of Emergency and Organ Transplantation, University of Bari, 70121 Bari, Italy; alessandro.bortone@gmail.com; 6Cardiology Unit, San Camillo Hospital, 00152 Rome, Italy; muguccioni@hotmail.com

**Keywords:** atrial fibrillation, DOACs, diabetes mellitus, risk index

## Abstract

Diabetes mellitus (DM) represents an independent risk factor for chronic AF and is associated with unfavorable outcomes. We aimed to evaluate the efficacy and safety of direct oral anticoagulants (DOACs) in patients with atrial fibrillation (AF), with and without diabetes mellitus (DM), using a new risk index (RI) defined as: RI $=\frac{\mathrm{Rate}\;\mathrm{of}\;\mathrm{Events}}{\mathrm{Rate}\;\mathrm{of}\;\mathrm{Patients}\;\mathrm{at}\;\mathrm{Risk}}$. In particular, an RI lower than 1 suggests a favorable treatment effect. We searched MEDLINE, MEDLINE In-Process, EMBASE, PubMed, and the Cochrane Central Register of Controlled Trials. The risk index (RI) was calculated in terms of efficacy (rate of stroke/systemic embolism (stroke SEE)/rate of patients with and without DM; rate of cardiovascular death/rate of patients with and without DM) and safety (rate of major bleeding/rate of patients with and without DM) outcomes. AF patients with DM (n = 22,057) and 49,596 without DM were considered from pivotal trials. DM doubles the risk index for stroke/SEE, major bleeding (MB), and cardiovascular (CV) death. The RI for stroke/SEE, MB, and CV death was comparable in patients treated with warfarin or DOACs. The lowest RI was in DM patients treated with Rivaroxaban (stroke/SEE, RI = 0.08; CV death, RI = 0.13). The RIs for bleeding were higher in DM patients treated with Dabigatran (RI110 = 0.32; RI150 = 0.40). Our study is the first to use RI to homogenize the efficacy and safety data reported in the DOACs pivotal studies against warfarin in patients with and without DM. Anticoagulation therapy is effective and safe in DM patients. DOACs appear to have a better efficacy and safety profile than warfarin. The use of DOACs is a reasonable alternative to vitamin-K antagonists in AF patients with DM. The RI can be a reasonable tool to help clinicians choose between DOACs or warfarin in the peculiar set of AF patients with DM.

## 1. Introduction

Atrial fibrillation (AF) is the most common arrhythmia worldwide. The prevalence of AF is expected to increase 2.5-fold in the next 50 years due to the growing mean age of the population [1]. Diabetes can be considered a pandemic too [2,3]. Diabetes mellitus (DM) represents an independent risk factor for chronic AF [4]. The development of AF is likely to be multifactorial and the mechanism is elusive, while evidence is emerging on the correlation between AF and DM [4]. DM and AF certainly share common risk factors, including hypertension, dyslipidemia, atherosclerosis, and obesity. Population-based studies suggested that DM is an independent risk factor for atrial fibrillation [5]. In patients with hypertension, DM did not act as an independent predictor for new onset AF in a post-hoc analysis from ALLHAT [6]. Nevertheless, a retrospective analysis of the VALUE study showed that hypertensive patients with new onset DM had a significantly higher event rate of new onset AF compared to patients without DM, even after adjusting for body mass index [7]. On the other hand, DM is one of the most common concomitant diseases in patients with AF [8]. Indeed, DM and AF are both predictors for stroke and mortality [9].

DM itself is associated with increased thrombin production and consequently may increase thromboembolic risk [10,11]. Anticoagulation therapy is mandatory in DM patients with AF. The use of VKA in these patients is to be implemented with caution. Hyperglycemia induces an increase in glycated albumin in DM patients. Glycated albumin has a reduced binding affinity for warfarin, resulting in a higher free fraction of the anticoagulant [12]. Consequently, there is a greater variability of the INR in AF patients with an increased risk of Stroke/SEE and MB [13].

Prevention of thromboembolic events was improved with the use of direct oral anticoagulants (DOACs) (Dabigatran, Rivaroxaban, Apixaban, and Edoxaban), which overcame the limitations of therapeutic standard of dose-adjusted vitamin K antagonists (VKAs) [14,15,16,17,18]. These drugs were approved based on the results from their respective dose-adjusted phase III, warfarin-controlled, randomized controlled trials (RCTs) [14,15,16,17].

The proportion of patients with DM enrolled in the four trials was 23% in the Randomized Evaluation of Long-Term Anticoagulant Therapy (RE-LY) study, 40% in the Rivaroxaban Once-daily oral direct factor Xa inhibition Compared with vitamin K antagonism for prevention of stroke and Embolism Trial in Atrial Fibrillation (ROCKET-AF) study, 25% in the Apixaban for Reduction in Stroke and Other Thromboembolic Events in Atrial Fibrillation (ARISTOTLE) study, and 36% in the Effective Anticoagulation With Factor Xa Next Generation in Atrial Fibrillation (ENGAGE-AF TIMI 48) study [14,15,16,17].

This systematic review aimed at evaluating the efficacy and safety of DOACs versus warfarin in patients with AF, with and without DM, by applying the risk index (RI) proposed by Uguccioni et al. [19,20].

## 2. Methods

We performed an extensive literature search to identify studies reporting stroke and systemic embolism, major bleeding, and cardiovascular (CV) death in patients with AF, randomized to VKA or DOAC, with and without DM. The search was performed in MEDLINE, MEDLINE In-Process, and Other Non-Indexed Citations, EMBASE, PubMed, and the Cochrane Central Register of Controlled Trials through the Ovid interface to identify English-language clinical articles published from 2002 (first marketed DOAC) to February 2020 related to phase III RCTs of dabigatran, rivaroxaban, apixaban, or edoxaban versus warfarin for the prevention of thrombotic events in AF patients. Keywords used were: “atrial fibrillation”, “warfarin”, “oral thrombin inhibitor”, “oral factor Xa inhibitor”, “dabigatran”, “rivaroxaban”, “apixaban”, “edoxaban”, and “diabetes”.

We also established regular alerts and complemented the electronic search strategy with a direct, manual reference review.

Systematic reviews, which included RCTs that evaluated stroke/systemic embolic events (SEE), major bleeding, and/or cardiovascular (CV) death and evaluated DOACs and VKAs were eligible for inclusion. PICOS (patients, intervention, comparator, outcomes, and study design) criteria for inclusion and exclusion of network meta-analyses (NMAs) are described in Table 1.

The search results were compared and the duplicates eliminated. An initial screening of the studies was performed on the basis of titles and abstracts, and then the full texts were reviewed by five. Five reviewers (D.A., P.S., C.A., V.C., and G.C.) independently performed the revision, while discrepancies were solved by a consensus, involving two additional authors (M.U., M.M.C.).

Data were derived from four pivotal trials (Figure 1). Details of the search strategy according to PRISMA-P were described in all of the tables in the Appendix A section.

The RI was computed in terms of efficacy (rate of stroke-systemic embolism/rate of patients with and without diabetes) and safety (rate of major bleeding/rate of patients with and without diabetes; rate of cardiovascular death/rate of patients with and without diabetes) of DOACs and VKAs.

## 3. Statistical Analysis

No statistical analyses were conducted; according to the authors, indirect, comparative meta-analyses among DOACs are a hypothesis generator that cannot provide definitive answers.

Furthermore, the RI does not allow the comparisons of rates among non-homogeneous studies.

## 4. Main Results

The main characteristics of the four RCTs involving DOACs are summarized in Table 2. About 22,057 patients with AF and DM and 49,596 AF patients without DM were finally included. The results of our systematic review are summarized in Table 3. The percentages of patients with DM ranged from 23.3% to 39.9%. The highest number of patients with DM was in the patient population treated with Rivaroxaban (40.3%).

Table 4 summarized data regarding the rate of stroke/SEE, major bleeding, and CV death related to warfarin, Dabigatran 110 mg and 150 mg BID, Rivaroxaban 20 mg QD, Apixaban 5 mg BID, and Edoxaban high and low dose (60−30 mg) QD.

The RIs for stroke/SEE and CV death were similar between patients treated with DOACs and patients treated with warfarin (Figure 2 and Figure 3, Table 5), except for Edoxaban 30 mg QD, which showed a higher RI than warfarin for stroke/SEE. Indeed, Rivaroxaban QD had the lowest RI values in terms of both stroke/SEE and CV death (Figure 2 and Figure 3, Table 5). Nevertheless, no data about CV death were reported for Edoxaban low doses, as none of the patients on Edoxaban low dose were included in the pivotal RCT (Table 5).

The RIs for major bleedings are shown in Table 5 and Figure 4. Dabigatran 150 mg BID showed a higher risk for bleeding compared to warfarin, while other DOACs showed substantially equal RIs in comparison with warfarin (Table 5 and Figure 4). No DM patients with AF as compared to DM patients without AF showed lower RIs (Table 5, Figure 2, Figure 3 and Figure 4).

## 5. Discussion

Diabetes mellitus per se increases the risk of systemic stroke/embolism or cardiovascular death in patients with atrial fibrillation [21]. Therefore, anticoagulant therapy is mandatory in these patients [22]. Choosing the correct anticoagulant therapy in patients with AF is a challenge, especially in patients at higher risk such as DM patients. DM and AF are two sides of the same coin due to the tight correlation between these two pathological conditions [4]. The frequency of NVAF increases by 40% in patients with Type 2 diabetes, while thromboembolic risk is 79% higher compared to non-diabetic patients [14,22,23,24]. Indeed, both conditions can potentiate their negative effects on systemic tissue and organs: for example, they can both promote kidney failure [25,26]. In such a setting, choosing the correct anticoagulation remains challenging as these drugs may also promote systemic alterations. Warfarin, the traditional anticoagulant used to prevent thromboembolism, can also be dangerous due to a supposed negative influence on kidneys and systemic vessels by promoting arterial calcification and decreasing renal function [27]. Indeed, DOACs seemed to be safer and more efficacious [27], especially when dealing with Type 2 diabetic patient [22,23,24].

Using RI as suggested by Uguccioni et al. could help reduce heterogeneity enrolled in RCTs and provide a better approach for the selection of the correct anticoagulant based on the different patient characteristics [19,20]. In particular, an RI lower than 1 suggests a favorable treatment effect. The lower the RI value, the better the performance of the drug within the specific context [19,20]. This is the first report that evaluates the risk for stroke/SEE, CV death, and major bleeding in patients with AF with and without DM by means of RI.

In our study, both DOACs and warfarin appear to be effective in preventing stroke and systemic embolism, with lower rates of CV death and major bleeding. The RI of each drug is lower than one, although some differences should be outlined. In particular, Dabigatran 150 mg BID might increase major bleeding risk as compared to warfarin, while the risk for stroke/SEE seems higher with Edoxaban 30 mg QD as compared to warfarin in DM patients. These data are the most contradictory in the panel of RIs comparisons as all of the other drugs and dosages revealed protective and efficient effects on patients’ outcomes as compared to warfarin (Figure 2 and Figure 4). By considering the absolute measurements, Rivaroxaban QD demonstrated the lowest RI value in terms of stroke/SEE outcomes, while Edoxaban 30 mg QD showed the lowest RI in terms of major bleeding outcomes, although no data are reported on pivotal Edoxaban Low Dose RCT with regard to CV death in patients with and without DM.

The literature offers evidence about the DOACs performances in AF patients in terms of efficacy and safety, but the reproducibility of the data and indirect comparisons among drugs may interfere with the correct choice of anticoagulants [28]. A meta-analysis by Ruff et al. involving the 71,683 patients with AF from registration RCTs showed a significant reduction in the incidence of stroke and SEE (relative risk (RR) 0.81, confidence interval (CI) 95% 0.73–0.91; *p* < 0.0001) in patients with DOACs versus warfarin, as well as all-cause mortality (RR 0.90, CI 95% 0.85–0.95; *p* = 0.0003) and intracranial hemorrhages (RR 0.48, 95% CI 0.39–0:59; *p* < 0.0001), despite the increase in gastrointestinal bleeding (RR 1.25, 95% CI 1.01–1.55; *p* = 0.04) [18].

## 6. Study Limitations

Unfortunately, a number of methodological discrepancies, including study design, different selection of the populations, and different definitions of outcomes among the four phase-III RCTs, reduce the generalization of results and the comparisons among drugs [14,15,16,17]. The selection of DM patients among RCTs populations aimed at evaluating a uniform subset of individuals.

Registration studies differ in terms of thromboembolic risk of the enrolled populations, age, heart failure, and active cancer. Active cancer is a high thromboembolic risk condition, and DAOCs appear to be an effective and safe therapeutic option in these patients [29]. The highest rate of DM patients was in Rocket-AF (40%) while ARISTOTLE had 25% and RE-LY had 23%.

The incidence of major bleeding was similar in AF patients with DM treated with Rivaroxaban or warfarin, while increasing when Dabigatran 110 mg, Dabigatran 150 mg, or warfarin were adopted. This difference might be associated with patients’ risk profiles, as well as other factors that may influence the pharmacokinetics and pharmacodynamics (co-morbidities, advanced age, HF, diabetes, hepatic or renal insufficiency). RCTs did not provide routine information about blood glucose levels and HbA_1c_, while patients with creatinine clearance < 30 mL/min were excluded. Therefore, patients with severe diabetic nephropathy, who are at even higher risk for cardiovascular complications, were excluded from RCTs. In these RCTs, no interactions were found between diabetic status and clinical efficacy of DOACs versus warfarin. Interestingly, the superiority of Apixaban over warfarin in terms of safety was lost when patients with AF and DM were considered (*p* = 0.003 for interaction) [30]. Moreover, a study-level meta-analysis in patients with DM and AF demonstrated a significant reduction in stroke and systemic embolism event rates (−20%) in patients treated with DOACs as compared to warfarin, as well as vascular mortality (−17%) and intracranial bleeding (−43%), while no influence was observed in terms of incidence of major bleeding [23].

We did not conduct any statistical analyses; indirect, comparative meta-analyses among DOACs are hypothesis generators and cannot provide definitive answers.

Conversely, it has been shown that published RCT data can be affected by the insertion of controversial data. In addition, they could invalidate the medical literature, alter the results of meta-analyses, and consequently compromise future research, political decisions, and above all patient care [31].

## 7. Conclusions

To our knowledge, no data are available about direct comparisons between DOACs in patients with DM. The choice of DOAC in patients with DM is not supported by specific evidence, but it should be guided by general principles, taking into account age and comorbidities (hypertension, coronary artery disease, heart failure, kidney disease, obesity, and dyslipidemia).

Diabetic patients show a doubled RI compared to non-diabetic patients. Data from the systematic evaluation of the four phase-III RCTs with DOACs in patients with AF and DM showed that Rivaroxaban had the lowest RI with regard to MB, CV death, and stroke/SEE. The use of DOACs is a reasonable alternative to VKAs in the management of patients with AF and DM. The risk index is a useful additional tool to help clinicians to choose DOACs or warfarin in a particular category of AF patients.

## Figures and Tables

**Figure 1 jcm-10-02924-f001:**
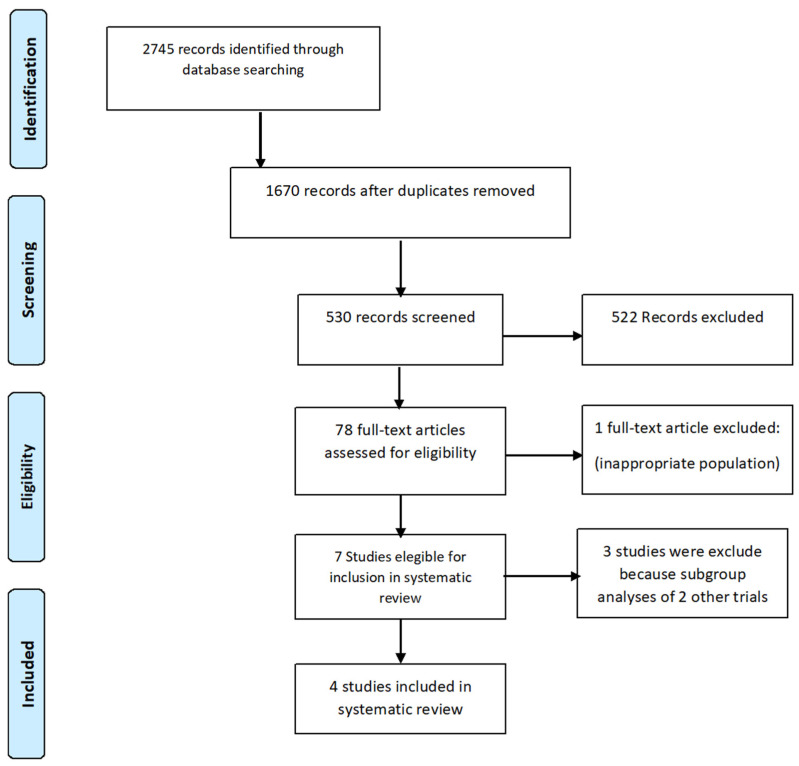
Preferred Reporting Items for Systematic Reviews and Meta-Analyses (PRISMA-P) flow diagram: search and selection process.

**Figure 2 jcm-10-02924-f002:**
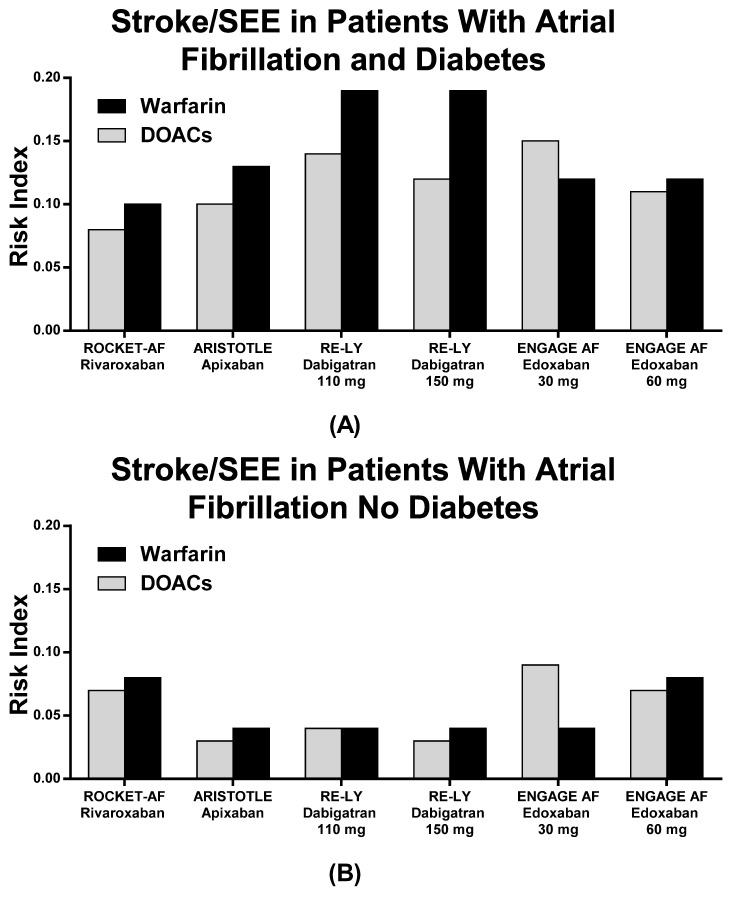
Risk index of stroke/systemic embolism in patients with (**A**) and without (**B**) diabetes in the pivotal trials.

**Figure 3 jcm-10-02924-f003:**
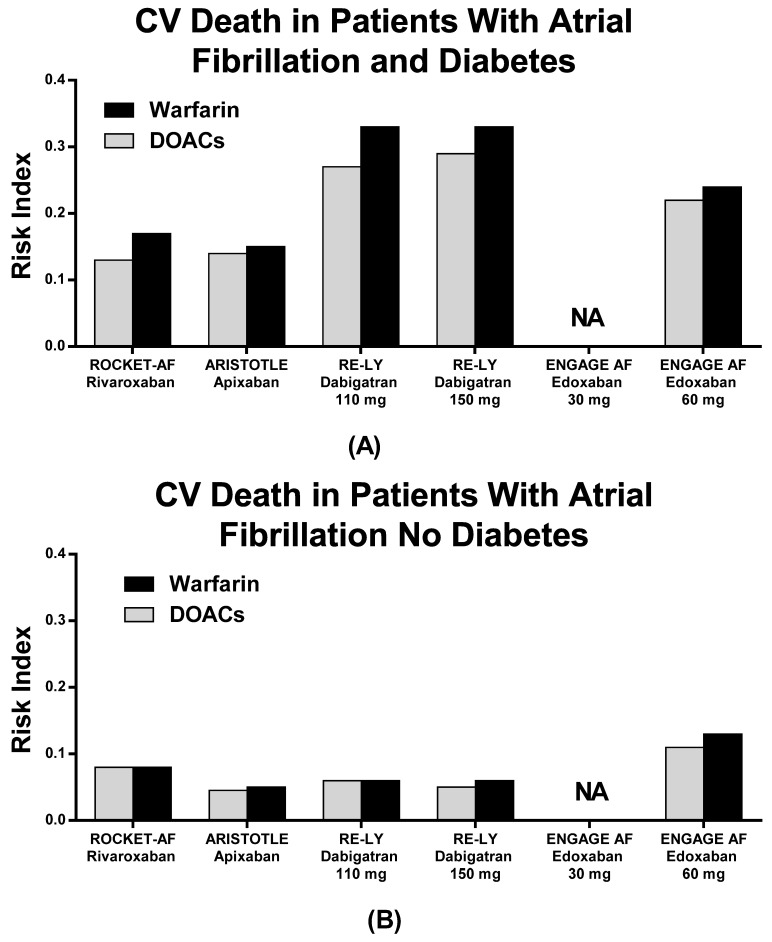
Risk index of CV Death in patients with (**A**) and without (**B**) diabetes in the pivotal trials.

**Figure 4 jcm-10-02924-f004:**
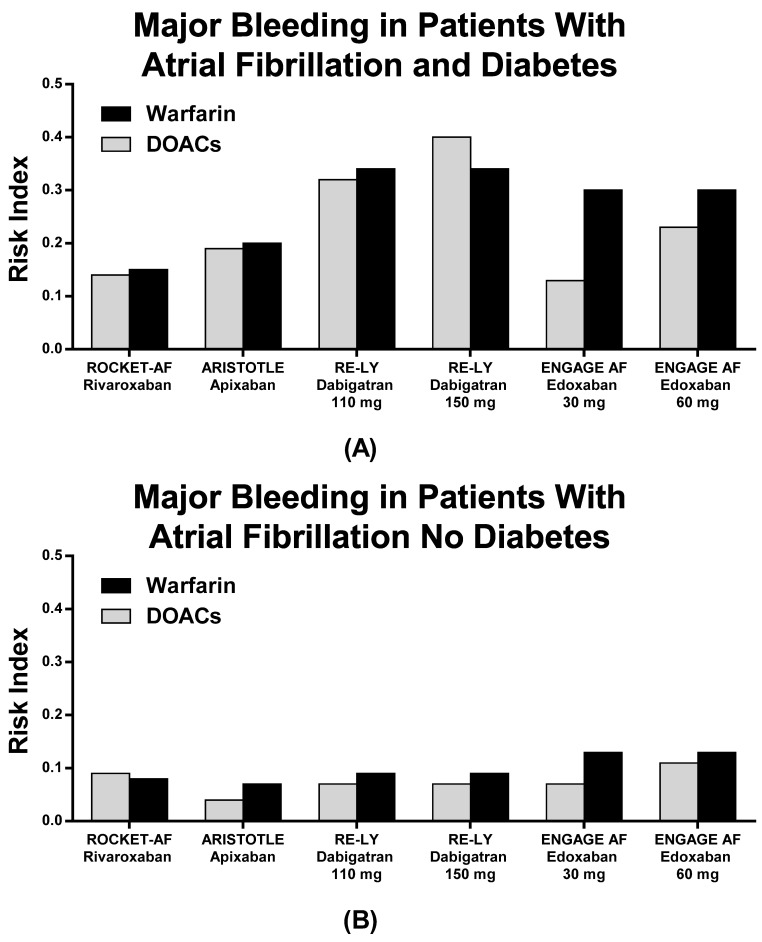
Risk index of Major Bleeding in patients with (**A**) and without (**B**) diabetes in the pivotal trials.

**Table 1 jcm-10-02924-t001:** PICOS criteria for inclusion and exclusion of systematic review.

	Inclusion Criteria	Exclusion Criteria
Population	Patients with NVAF receiving any of the treatments below. All studies in the SLR must include ≥ 90% patients with NVAF. SLRs including studies with < 90% patients with NVAF must report data separately for the NVAF studies	Not a population of interest (ie, non-NVAF patients) Studies of patients receiving ablation, cardioversion, or left-atrial appendage closure
Intervention/comparator	DOACs (apixaban, dabigatran, rivaroxaban, edoxaban) and warfarin studies need to have compared 1 or more DOACs and/or warfarin	Studies not reporting outcomes for population of interest
Outcome	Clinical outcomes: ▪ Stroke/systemic embolism ▪ Major bleeding (ISTH or modified ISTH). ▪ Cardiovascular death ▪ Patients with and without diabetes mellitus Doses included: Apixaban: 5 or 2.5 mg ^a^ Rivaroxaban: 20 or 15 mg Dabigatran: 150 or 110 mg Edoxaban: 30 or 60 mg	SLRs/NMAs of observational studies, nonsystematic reviews, primary research trials, primary observational studies, case reports, case series, narrative reviews Letters to the editor, guidelines, meeting abstracts In vitro pharmacodynamic or pharmacokinetic studies only, animal studies, genetic studies only
Study design	SLR of randomized controlled trials	

PICOS, patients, intervention, comparator, outcomes, study design; DOAC, direct oral anticoagulant; ISTH, International Society on Thrombosis and Hemostasis; CHADS_2_ = (congestive heart failure, hypertension, age ≥ 75 years), diabetes mellitus, stroke (double weight)); NMA, network meta-analysis; NVAF, nonvalvular atrial fibrillation; SLR, systematic literature review. ^a^ Any network meta-analysis comparison of apixaban 2.5 mg only with another DOAC was not included.

**Table 2 jcm-10-02924-t002:** Main characteristics of the four DOACs pivotal trials.

	ROCKET AF Rivaroxaban	ARISTOTLE Apixaban	RE-LY Dabigatran	ENGAGE Edoxaban
Effect	Anti-Xa	Anti-Xa	Anti-IIa	Anti-Xa
Dose	20/15 mg QD	5/2.5 mg BID	150/110 mg BID	60/30 mg QD
Mean CHADS_2_ score	3.5	2.1	2.1	2.8
Target INR (Warfarin arm)	2–3	2–3	2–3	2–3
TTR (%)	58	62	64	68
Asia Pacific Region N (%)	2109 (14.8%)	2916 (16%)	3854 (21%)	3383 (16%)
Median Follow-up duration	1.9 y	1.8 y	2 y	2.8 y

ROCKET-AF = Rivaroxaban Once-daily oral direct factor Xa inhibition Compared with vitamin K antagonism for prevention of stroke and Embolism Trial in Atrial Fibrillation [17]; ARISTOTLE = Apixaban for Reduction in Stroke and Other Thromboembolic Events in Atrial Fibrillation [16]; RE-LY = Randomized Evaluation of Long-Term Anticoagulant Therapy [14]; ENGAGE-AF TIMI 48 = Effective Anticoagulation With Factor Xa Next Generation in Atrial Fibrillation [15]; Xa = Factor X activated; IIa = Thrombin; QD = Quaque die; BID = Bis in die; CHADS_2_ = Congestive Heart Failure, Hypertension, Age, Diabetes, Stroke; INR = International Normalized Ratio; TTR = Time to Range; N = Number; y = year.

**Table 3 jcm-10-02924-t003:** Patients with and without diabetes in the pivotal trials.

RCTs	Pts on DOACs (N)	Diabetes N (%)	No Diabetes N (%)	Pts on Warfarin (N)	Diabetes N (%)	No Diabetes N (%)
ROCKET AF	7131	2878 (40.3%)	4253 (59.7%)	7133	2817 (39.5%)	4316 (60.5%)
ARISTOTLE	9120	2284 (25.0%)	6836 (75%)	9087	2263 (24.9%)	6818 (75.1%)
RE-LY _110 mg_	6015	1409 (23.4%)	4606 (66.6%)	6022	1410 (23.4%)	4612 (66.6%)
RE-LY _150 mg_	6076	1402 (23.1%)	4674 (63.9%)	6022	1410 (23.4%)	4612 (66.6%)
ENGAGE _30 mg_	7034	2544 (36.2%)	4490 (63.8%)	7036	2521 (35.8%)	4515 (64.2%)
ENGAGE _60 mg_	7035	2529 (35.9%)	4476 (64.1%)	7036	2521 (35.8%)	4515 (64.2%)

RCTs = randomized controlled trials; DOACs = non-VKA antagonist drugs; Pts = Patients; N = Number; ROCKET-AF = Rivaroxaban Once-daily oral direct factor Xa inhibition Compared with vitamin K antagonism for prevention of stroke and Embolism Trial in Atrial Fibrillation [17]; ARISTOTLE = Apixaban for Reduction in Stroke and Other Thromboembolic Events in Atrial Fibrillation [16]; RE-LY = Randomized Evaluation of Long-Term Anticoagulant Therapy [14]; ENGAGE-AF TIMI 48 = Effective Anticoagulation With Factor Xa Next Generation in Atrial Fibrillation [15].

**Table 4 jcm-10-02924-t004:** Stroke/systemic embolism, major bleeding, and cardiovascular death in patients with and without diabetes in the pivotal trials.

**Diabetes**						
RCTs	Stroke o SEE (N)%		MB (N)%		CV Death N (%)	
	DOACs	Warfarin	DOACs	Warfarin	DOACs	Warfarin
ROCKET AF	95 (3.3%)	114 (4.0%)	165 (5.7%)	169 (6.0%)	152 (5.3%)	192 (6.8%)
ARISTOTLE	57 (2.5%)	75 (3.3%)	112 (4.9%)	114 (5.0%)	79 (3.5%)	88 (3.9%)
RE-LY _110_	49 (3.5%)	64 (4.5%)	106 (7.5%)	114 (8.0%)	91 (6.5%)	109 (7.7%)
RE-LY _150_	40 (2.9%)		129 (9.2%)		95 (6.8%)	
ENGAGE _30 mg_	135 (5.3%)	107 (4.2%)	123 (4.9%)	278 (11%)	NA	NA
ENGAGE _60 mg_	102 (4.0%)		219 (8.6%)		209 (8.1%)	219 (8.6%)
**No Diabetes**						
RCTs	Stroke o SEE (N)%		MB (N)%		CV Death N (%)	
ROCKET AF	174 (4.0%)	192 (4.4%)	230 (5.4%)	217(5.0%)	223 (5.2%)	209 (4.8%)
ARISTOTLE	155 (2.3%)	190 (2.8%)	215 (3.2%)	348 (5.1%)	229 (3.4%)	256 (3.7%)
RE-LY _110_	134 (2.9%)	138 (2.9%)	236 (5.1%)	307 (6.6%)	198 (4.3%)	208 (4.5%)
RE-LY _150_	94 (2.0%)		271 (5.8%)		179 (3.8%)	
ENGAGE _30 mg_	NA	NA	NA	NA	NA	NA
ENGAGE _60 mg_	207 (4.6%)	248 (5.4%)	335 (7.5%)	417 (9.2%)	331 (7.4%)	406 (8.9%)

RCTs = randomized controlled trials; DOAC = non-VKA antagonist drugs; N = Number; SEE = systemic embolism; MB = major bleeding; CV = cardiovascular death; ROCKET-AF = Rivaroxaban Once-daily oral direct factor Xa inhibition Compared with vitamin K antagonism for prevention of stroke and Embolism Trial in Atrial Fibrillation [17]; ARISTOTLE = Apixaban for Reduction in Stroke and Other Thromboembolic Events in Atrial Fibrillation [16]; RE-LY = Randomized Evaluation of Long-Term Anticoagulant Therapy [14]; ENGAGE-AF TIMI 48 = Effective Anticoagulation With Factor Xa Next Generation in Atrial Fibrillation [15].

**Table 5 jcm-10-02924-t005:** Risk index of stroke/systemic embolism, major bleeding, and cardiovascular death in patients with and without diabetes in the pivotal trials.

**Diabetes**						
RCTs	Stroke o SEE		MB		CV Death	
	DOACs	Warfarin	DOACs	Warfarin	DOACs	Warfarin
ROCKET AF	0.08	0.10	0.14	0.15	0.13	0.17
ARISTOTLE	0.10	0.13	0.19	0.20	0.14	0.15
RE-LY _110_	0.14	0.19	0.32	0.34	0.27	0.33
RE-LY _150_	0.12		0.40		0.29	
ENGAGE _30 mg_	0.15	0.12	0.13	0.30	NA	NA
ENGAGE _60 mg_	0.11		0.23		0.22	0.24
**No Diabetes**						
RCTs	Stroke o SEE		MB		CV Death	
ROCKET AF	0.06	0.07	0.09	0.08	0.08	0.08
ARISTOTLE	0.03	0.03	0.04	0.06	0.04	0.04
RE-LY _110_	0.04	0.04	0.07	0.09	0.06	0.06
RE-LY _150_	0.03		0.09		0.06	
ENGAGE _30 mg_	NA	NA	NA	NA	NA	NA
ENGAGE _60 mg_	0.07	0.08	0.11	0.14	0.11	0.13

RCTs = randomized controlled trials; N = Number; SEE = systemic embolism; MB = major bleeding; CV = cardiovascular death; DOAC = non-VKA antagonist drugs; ROCKET-AF = Rivaroxaban Once-daily oral direct factor Xa inhibition Compared with vitamin K antagonism for prevention of stroke and Embolism Trial in Atrial Fibrillation [17]; ARISTOTLE = Apixaban for Reduction in Stroke and Other Thromboembolic Events in Atrial Fibrillation [16]; RE-LY = Randomized Evaluation of Long-Term Anticoagulant Therapy [14]; ENGAGE-AF TIMI 48 = Effective Anticoagulation With Factor Xa Next Generation in Atrial Fibrillation [15].

## Data Availability

Data are available on request by contacting the corresponding author.

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
