# Peer review of "A Systematic Review of the Efficacy and Safety of Direct Oral Anticoagulants in Atrial Fibrillation Patients with Diabetes Using a Risk Index"

_jcm, 2021, doi:10.3390/jcm10132924_

Round 1
Reviewer 1 Report
This review untitled “A systematic Review of the Efficacy and Safety of Direct Oral Anticoagulants in Atrial Fibrillation Patients with Diabetes using a Risk Index.” by Domenico Acanfora et al. provides an interesting and comprehensive evaluation of the efficacy and safety of Direct Oral Anticoagulants (DOACs) in patients with atrial fibrillation (AF), with Diabetes Mellitus (DM) using a new risk index (RI).
However several points are not clear and an appropriate revision is highly demanded.
- Introduction: warfarin and VKA are just mentioned without any information regarding their use in clinical practice in DM patients. However, authors related the effects of DOACs to warfarin so a clear and comprehensive explaination has to be added.
- Why VKAs are not take in consideration regarding their effects in all the paramenters mentioned?
- I guess you have to add in the figures also the effects of VKAs.
- How can you statistically identify that Rivoraxaban treatment is better than Warfarin and the other DOACs??.
Major revisions. I suggest to revise the paper, as I suggested, emphasizing the points written above.
Author Response
Reviewer #1
We thank this Reviewer for the constructive comments and suggestions. Furthermore, we would like to really thank him/her for his/her appreciation about our research in the introduction section of his/her comments. This is our point-to-point reply.
Introduction: warfarin and VKA are just mentioned without any information regarding their use in clinical practice in DM patients. However, authors related the effects of DOACs to warfarin so a clear and comprehensive explanation has to be added.
Done. The introduction section was re-structured. We added “DM itself is associated with increased thrombin production and consequently may increase thromboembolic risk [10,11]. Anticoagulation therapy is mandatory in DM patients with AF. The use of VKA in these patients is to be implemented with caution. Hyperglycemia induces an increase in glycated albumin in DM patients. Glycated albumin has a reduced binding affinity for warfarin, resulting in a higher free fraction of the anticoagulant [12]. Consequently a greater variability of the INR in AF patients with an increased risk of Stroke/SEE and MB [13].” We added references 12 and 13.
Why VKAs are not take in consideration regarding their effects in all the paramenters mentioned?
Done. In the main results section, we added Warfarin “Table 4 summarized data regarding the rate of stroke/SEE, major bleeding, and CV death related to Warfarin, Dabigatran 110 mg and 150 mg BID, Rivaroxaban 20 mg QD, Apixaban 5 mg BID, and Edoxaban high and low dose (60-30 mg) QD.”
I guess you have to add in the figures also the effects of VKAs.
Done.
How can you statistically identify that Rivoraxaban treatment is better than Warfarin and the other DOACs??
We have not reported the RI for Rivaroxaban as statistical, but only as descriptive.
Reviewer 2 Report
Selecting the appropriate anticoagulation therapy in patients with AF is a significant challenge, especially in subjects at higher risk, such as patients with DM. Therefore, the analysis performed with the use of the RI index, based on the results of large pivotal studies on patients with AF and DM, is particularly valuable and helpful. The results are convincing and quite interesting. Still, I would like to make some remarks:
1 / The analysis data comes from four pivotal studies, however, due to a large number of methodological discrepancies between these clinical trials, including study design, different selection of the populations, and different definitions of outcomes , the results are not based on statistical analysis.
2/Indirect comparative meta-analyzes among patients treated with DOACs and warfarin allow only to create hypotheses on the basis of which it is impossible to draw unambiguous conclusions
3/Although the authors mention most of the shortcomings of the analysis, I believe that the paragraph devoted to these issues should be specified separately in the discussion in the form of the „study limitation”.
4/ The sentence used in the discussion: "The highest rate of DM patients was in ROCKET-AF (40%) while only a half of ARISTOTLE and RE-LY patients had DM (25% and 23%, respectively)", is inaccurate as 25% and 23% of DM patients did not account for half of the patients in these studies.
5/The problem of choosing an anticoagulant in patients with AF and DM is difficult, as in patients with AF and cancer, for whom the use of DOAC also appears to be effective and safe. Therefore, the paper would benefit by briefly addressing this topic and including the reference: Int J Cardiol. 2021 Feb 1;324:78-83. doi: 10.1016/j.ijcard.2020.09.037. Epub 2020 Sep 12. PMID: 32931852
Author Response
Reviewer #2
We thank this Reviewer for the constructive comments and suggestions. Furthermore, we would like to really thank him/her for his/her appreciation about our research in the introduction section of his/her comments. This is our point-to-point reply.
The analysis data comes from four pivotal studies, however, due to a large number of methodological discrepancies between these clinical trials, including study design, different selection of the populations, and different definitions of outcomes , the results are not based on statistical analysis.
I agree. Some differences between the trials are reported in the text and in Table 2.
Indirect comparative meta-analyzes among patients treated with DOACs and warfarin allow only to create hypotheses on the basis of which it is impossible to draw unambiguous conclusions.
I agree.
Although the authors mention most of the shortcomings of the analysis, I believe that the paragraph devoted to these issues should be specified separately in the discussion in the form of the study limitation.
Done.
The sentence used in the discussion: "The highest rate of DM patients was in ROCKET-AF (40%) while only a half of ARISTOTLE and RE-LY patients had DM (25% and 23%, respectively)", is inaccurate as 25% and 23% of DM patients did not account for half of the patients in these studies.
Done. We changed the sentence as follows “The highest rate of DM patients was in Rocket-AF (40%) while ARISTOTLE had 25% and RE-LY had 23%.”
The problem of choosing an anticoagulant in patients with AF and DM is difficult, as in patients with AF and cancer, for whom the use of DOAC also appears to be effective and safe. Therefore, the paper would benefit by briefly addressing this topic and including the reference: Int J Cardiol. 2021 Feb 1;324:78-83. doi: 10.1016/j.ijcard.2020.09.037. Epub 2020 Sep 12. PMID: 32931852
Done. We have added the reference and changed the sentence as follows: Registration studies differ in terms of thromboembolic risk of the enrolled populations, age, heart failure, active cancer. Active cancer is a high thromboembolic risk condition and DAOCs appear to be an effective and safe therapeutic option in these patients. [29]

This manuscript is a resubmission of an earlier submission. The following is a list of the peer review reports and author responses from that submission.